# Relationship between ^18^F-FDG Uptake in the Oral Cavity, Recent Dental Treatments, and Oral Inflammation or Infection: A Retrospective Study of Patients with Suspected Endocarditis

**DOI:** 10.3390/diagnostics10090625

**Published:** 2020-08-24

**Authors:** Geertruida W. Dijkstra, Andor W. J. M. Glaudemans, Paola A. Erba, Marjan Wouthuyzen-Bakker, Bhanu Sinha, David Vállez García, Luc W. M. van der Sluis, Riemer H. J. A. Slart

**Affiliations:** 1Center for Dentistry and Oral Hygiene, University Medical Center Groningen, University of Groningen, PO 9700 RB Groningen, The Netherlands; g.w.dijkstra@umcg.nl (G.W.D.); l.w.m.van.der.sluis@umcg.nl (L.W.M.v.d.S.); 2Department of Nuclear Medicine and Molecular Imaging, University Medical Center Groningen, University of Groningen, PO 9700 RB Groningen, The Netherlands; paola.erba@unipi.it (P.A.E.); d.vallez-garcia@umcg.nl (D.V.G.); r.h.j.a.slart@umcg.nl (R.H.J.A.S.); 3Department of Nuclear Medicine, Department of Translational Research and New Technology in Medicine, University of Pisa, 56128 Pisa, Italy; 4Department of Medical Microbiology and Infection Prevention, University Medical Center Groningen, University of Groningen, PO 9700 RB Groningen, The Netherlands; m.wouthuyzen-bakker@umcg.nl (M.W.-B.); b.sinha@umcg.nl (B.S.); 5Department of Biomedical Photonic Imaging, University of Twente, 7522 NB Enschede, The Netherlands

**Keywords:** infective endocarditis, oral health, FDG PET/CT

## Abstract

[^18^F]-fluorodeoxyglucose positron emission tomography ([^18^F]FDG PET/CT) has proven to be a useful diagnostic tool in patients with suspected infective endocarditis (IE), but is conflicting in relation to dental procedures. Questions: Is there a correlation between [^18^F]FDG PET/CT findings, recent dental treatment, and an affected oral cavity? (2) Is there a correlation between infective endocarditis (IE), oral health status, and (extra)cardiac findings on [^18^F]FDG PET/CT? Methods: This retrospective study included 52 patients. All [^18^F]FDG PET/CT scans were examined visually by pattern recognition using a three-point scale and semi-quantified within the volume of interest (VOI) using SUV_max_. Results: 19 patients were diagnosed with IE (group 1), 14 with possible IE (group 2), and 19 without IE based on the modified Duke criteria (group 3). No correlation was found between visual PET and SUV_max_ and sites of oral inflammation and infection. The visual PET scores and SUV_max_ were not significantly different between all groups. A significant difference in the SUV_max_ of the valve between all groups was observed. Conclusions: This study suggests that no correlation exists between the PET findings in the oral cavity and dental treatments or inflammation/infection. No correlation between IE, actual oral health status, and extra-cardiac findings was demonstrated. Additional research is needed to conclude whether [^18^F]FDG PET/CT imaging is a reliable diagnostic modality for oral inflammation and infection sites.

## 1. Introduction

The oral cavity is a natural habitat for more than 500 species of microorganisms [1,2]. These microorganisms can form biofilms on dental surfaces. In the case of an unbalance among the microbial species in the biofilm, oral diseases can emerge, such as caries and periodontitis, the most prevalent microbial diseases in humans [2]. Indeed, any disruption of the natural integrity between the biofilm and the subgingival epithelium, which is at most about 10 cell layers thick, could lead to a bacteremic state. Oral microorganisms can also enter the circulation and cause bacteremia during and after routine oral activities (e.g., chewing and tooth brushing) or dental procedures (e.g., dental extractions, scaling, and root planing) [3,4]. The presence of gingivitis, periodontitis, or other odontogenic infections (following dental procedures—e.g., extractions, dental scaling) increases the incidence of bacteremia, since the periodontal vasculature proliferates and dilates, providing an even greater surface area to allow the entry of microorganisms into the bloodstream [5]. This may lead to the seeding of organisms in different target organs, resulting in subclinical, acute, or chronic infections. Infective endocarditis (IE), a microbial infection of a native or prosthetic heart valve, the endocardial surface, other organs, or an indwelling cardiac device [6], is a well-known complication of odontogenic bacteremia [7]. IE is a serious condition with a 30% incidence of mortality [3,8,9].

The bacteria that cause IE are in 85% of cases of Gram-positive cocci (streptococci, staphylococci, and enterococci) that can be considered as commensals with *Staphylococcus aureus* (*S. aureus*), associated with the most severe form of IE [10]. The most prevalent species of the streptococci group causing IE is *Streptococcus viridans* (*S. viridans*) [10,11,12]. The blood-circulating bacteria responsible for IE enter the bloodstream via the mouth (29%, of which 59% have an oral or dental infectious focus, and 12% are due to recent dental procedures), the skin (40%), and the gastrointestinal and urinary tracts (23%) [13]. A higher microbial load would facilitate such dissemination, as it is known that individuals with poor oral hygiene are at higher risk of developing bacteremias during oral manipulative procedures [14].

In case of infection, [^18^F]-fluorodeoxyglucose positron emission tomography ([^18^F]FDG PET/CT) is used to provide diagnostic information based on increased glucose metabolism by inflammatory cells recruited at the site of infection [15,16,17]. In addition, Heuker et al. (2017) [17] observed that the clinical isolates of many major Gram-positive and Gram-negative bacterial pathogens can actively take up [^18^F]FDG. Since both types of bacteria are present in the oral cavity, [^18^F]FDG PET/CT can be used to detect periodontal disease and apical periodontitis [18,19,20]. Furthermore, a correlation was described between peri-implant hypermetabolism and peri-implantitis diagnosed in [^18^F]FDG PET/CT scans [21].

Whole-body [^18^F]FDG PET/CT has been demonstrated to successfully identify cardiac and extra-cardiac infection, providing information on possible portals of infection (POEs)—e.g., in the case of IE, the oral cavity [22,23]. According to the recently published guidelines of the European Society of Cardiology, [^18^F]FDG PET/CT should be a routine part of the diagnostic algorithm of patients with suspected IE [24]. However, no systematic analysis has examined the role of [^18^F]FDG PET/CT in detecting infection in the oral cavity in patients with IE. Therefore, this study aims (1) to explore the correlation of [^18^F]FDG uptake in patients with recent dental treatments and/or inflammation and infection in the oral cavity; (2) the correlation between IE, oral health status, and (extra)cardiac findings on [^18^F]FDG PET/CT.

## 2. Materials and Methods

### 2.1. Study Design and Patients

Between 2011 and 2018, 60 patients (39 men, 21 women; median age 60 years; IQR (48–69) years) with suspected IE underwent [^18^F]FDG PET/CT imaging at the University Medical Center Groningen (UMCG), the Netherlands. Status praesens and dental records were requested from the dental offices of the included patients. Exclusion criteria were patients who did not visit a dentist during the two years preceding the [^18^F]FDG PET/CT scan, and edentate patients without dental implants. All the patients were divided into three groups based on their final diagnosis: group 1—patients diagnosed with IE; group 2—patients diagnosed with possible IE; and group 3—patients without IE based on the modified Duke criteria [24]. The final diagnosis of IE was established through expert consensus based on all available clinical and diagnostic data, and with at least one year of follow-up. [^18^F]FDG PET/CT scans, transoesophageal echocardiograms, transthoracic echocardiograms, and blood cultures were analyzed for all the patients. The local institutional review board approved this retrospective study and waived the requirement for written informed consent (2015/033 (date: March 2016)).

### 2.2. Status Praesens, Oral Health, and Dental Records

Patient characteristics and dental records were reviewed for the relevant clinical data at the time of the [^18^F]FDG PET/CT scan, including age; sex; clinical risk factors; acute dental infection; decayed, missing, and filled teeth (DMFT index); and the status praesens of the oral cavity (number and sites of dental procedures). All the dental treatments such as dental restorations, extractions, implants, root canal treatments, non-surgical procedures (scaling and root planing), and surgical periodontal treatment within two years before the [^18^F]FDG PET/CT scan were assessed to verify whether a dental treatment could have caused the infection. The treatments were registered in six sextants (upper and lower jaw, incisor/canine region, left and right premolar/molar region) (Figure 1).

### 2.3. [^18^F]FDG PET/CT Acquisition and Interpretation

[^18^F]FDG PET/CT imaging was performed on a BioGraph 40- or 64-slice mCT (Siemens Healthcare, Knoxville, TN, USA), and reconstructions were performed according to the EANM-SNMMI guidelines and IE recommendations [24,25]. For all scans, the patients followed a 24 h low-carbohydrate/fat-allowed diet and fasted for a minimum of 6 h. All the patients were scanned from the skull base to mid-thigh. The [^18^F]FDG activity of 3 MBq/kg of body weight was injected intravenously 60 min before the PET data acquisition, according to existing guidelines [26]. All the scans were accompanied by low-dose CT scanning for attenuation correction and anatomical positioning and were performed early in the diagnostic process, within four days and no later than seven days after the start of antimicrobial therapy. All the scans were separately analyzed in consensus by two experienced readers (AWJMG, RHJAS), blinded for all clinical information. Image analysis was performed using Siemens Syngo.via (Client version 3.0; Siemens, Erlangen, Germany).

### 2.4. [^18^F]FDG PET/CT Scan Analyses of the Oral Cavity and Extra-Cardiac Areas

All the [^18^F]FDG PET/CT scans (with and without attenuation correction) were examined visually using a three-point scale per sextant: 0 = no [^18^F]FDG uptake visible; 1 = mildly increased uptake (equal to muscle background); 2 = moderately increased uptake (higher than muscle activity, lower than [^18^F]FDG brain activity); or 3 = highly increased uptake (equal or higher than brain activity). Uptake scores of 2 and 3 were defined as positive.

The [^18^F]FDG uptake was further evaluated qualitatively by pattern recognition (homogeneous or focal/heterogeneous). The [^18^F]FDG accumulation uptake was also semi-quantified by drawing a volume of interest (VOI) in all six sextants and calculating the SUV_max_ (Figure 1). Furthermore, the SUV_max_ was determined in the valve region. The valve culture was determined in patients who underwent a surgical valve replacement.

Extra-cardiac findings (lungs, mediastinum, sternum, CIED, liver, pancreas, gastrointestinal, lymphatic system, oropharynx, thyroid, musculoskeletal, abdominal, vascular, cutaneous, atrial, and sinuses) were measured as negative (0: no [^18^F]FDG uptake visible) or positive (1: [^18^F]FDG uptake visible).

### 2.5. Statistical Analyses

For continuous variables (age, SUV_max_, and the number of dental treatments), the normality of the distribution was visually explored using Q-Q-plots, histograms, and box-plots. Fisher’s exact probability test was used to analyze the visual [^18^F]FDG uptake scores for the oral cavity and dental treatments performed in 312 sextants in the two years before the [^18^F]FDG PET/CT scan. Furthermore, Pearson’s correlation was used to determine the relationship between the SUV_max_ values and the number of dental treatments.

[^18^F]FDG PET/CT was considered to be true-positive for oral findings when the SUV_max_ and visual scores per sextant matched with the dental treatments and true-negative when no [^18^F]FDG uptake was visually identified and no dental treatments were performed or disease was reported in the same sextant. [^18^F]FDG PET/CT was considered false-positive when positive findings were not related to any dental procedure or disease and false-negative in the case of negative PET/CT findings in patients with dental disease and if there were recent dental treatments.

To analyze a possible difference in the prevalence of dental treatments (0: no dental treatments performed; 1: dental treatments performed), Fisher’s exact probability test was conducted. The Kruskal–Wallis H test was used to compare the ordinal variable DMFT index within groups 1, 2, and 3, as well as the differences between PET/CT scores. A one-way ANOVA was conducted to compare the SUV_max_ measured in the oral cavity.

Furthermore, to analyze the difference between the SUV_max_ valve scores in groups 1, 2, and 3, a one-way ANOVA and Tukey’s post-hoc test were used. Pearson’s correlations were used to determine whether the oral visual [^18^F]FDG uptake scores were correlated with the presence of a visual uptake of [^18^F]FDG in a heart valve. Fisher’s exact probability test was used to analyze a possible difference in the prevalence of extra-cardiac [^18^F]FDG PET/CT findings between groups 1, 2, and 3.

*P* values < 0.05 were considered statistically significant. Statistical analyses were performed using the IBM Statistical Package for Social Sciences (SPSS), version 25 (IBM SPSS).

## 3. Results

Between 2011 and 2018, 60 consecutive [^18^F]FDG PET/CT scans were performed in patients with suspected IE. After the exclusion of six edentate patients and two patients who did not visit a dental professional in the two years before the PET/CT, 52 patients (32 men, 20 women; median age, 58 years; IQR (44–66) years) were included. Clinical parameters, including the echocardiography findings and blood cultures at the time of imaging, are summarized in Table 1. Of the 52 patients, 19 had a final diagnosis of IE (group 1: 12 men, 7 women; median age, 57 years; IQR (40–66) years), 14 had a final diagnosis of possible IE (group 2: 7 men, 7 women; median age, 60 years; IQR (37–65) years), and the remaining 19 patients were diagnosed as not having IE based on the modified Duke criteria (group 3: 13 men, 6 women; median age, 60 years; IQR (51–70) years).

### 3.1. Dental Procedures Prior to [^18^F]FDG PET/CT Imaging

In the two years before the [^18^F]FDG PET/CT scans, a total of 97 dental restorations, 43 extractions, 0 implants, and 5 root canal treatments were performed by 48 different dentists. The prevalence of patients who received dental restorations in groups 1, 2, and 3 was 47%, 86%, and 53%, respectively. The prevalence of patients who underwent extractions in groups 1, 2, and 3 was 21%, 7%, and 21%, respectively. For root canal treatments, the figures for groups 1, 2, and 3 were 16%, 7%, and 5%, respectively. No significant differences between groups were observed for the prevalence of dental restorations (*p* = 0.061), extractions (*p* = 0.590), or root canal treatments (*p* = 0.615).

Furthermore, all 52 patients routinely visited a dentist once or twice per year, which is the standard in the Netherlands. The removal of supragingival calculus by scaling was performed regularly in 17 patients (90%) in group 1, 11 patients (79%) in group 2, and 9 (47%) patients in group 3. The removal of subgingival calculus by scaling and root planing was performed in 1 patient (5%) in group 1, 0 patients in group 2, and 4 patients (21%) in group 3. None of the 52 patients had been subjected to periodontal surgery.

### 3.2. Oral Status Praesens at Time of [^18^F]FDG PET/CT Imaging

Fifty status praesens and DMFT indices (median 12; IQR [7.3–16.5]) were determined by the information obtained from dental records. A Kruskal–Wallis H test revealed no difference (H(2) = 4.724, *p* = 0.094) between the mean DMFT scores of the three groups. The DMFT score in group 1 (median 10.5; IQR [5,6,7,8,9,10,11,12,13,14]) was not significantly different than the DMFT score in group 3 (median 12.5; IQR [10.3–23.8]). Based on the dental records, the periodontal health was good in all patients. Only one patient had acute dental problems at the time of the scan (Figure 2).

### 3.3. Visual [^18^F]FDG Uptake Scores and SUV_max_ Compared with Dental Treatments

In 54 sextants (17.3%) the positive visual scores were between 2 and 3 (median 2.0) (Table 2). Dental restorations were performed in 69 sextants, of which only 13 had a positive visual score. Furthermore, extractions were performed in 16 sextants, of which only 2 had a positive visual score. All the root canal treatments were performed in sextants with a visual [^18^F]FDG uptake score of 0. Regarding the true-positive findings of [^18^F]FDG PET/CT scans, the number of dental treatments did not correlate with the visual [^18^F]FDG uptake scores (dental restorations, *p* = 0.546; extractions, *p* = 0.985; and root canal treatments, *p* = 1.000).

The SUV_max_ was measured in all sextants, and was above zero in 72 sextants (23%) (3.56 ± 1.67). The number of dental treatments also did not correlate with the SUV_max_ (dental restorations, *p* = 0.209; extractions, *p* = 0.933; root canal treatments, *p* = 0.593).

The visual [^18^F]FDG uptake scores in groups 1, 2, and 3 were not significantly different (H(2) = 4.359, *p* = 0.113). There was also no significant difference in the SUV_max_ for the three groups (F(2, 69) = 0.27, *p* = 0.798).

### 3.4. Cardiac and Extra-Cardiac [^18^F]FDG PET/CT Findings

There was a significant difference in the SUV_max_ of the valve for the three groups (F(2, 309) = 28.06, *p* < 0.001). A Tukey post-hoc test revealed that the SUV_max_ of the valve was significantly lower for group 2 (3.04 ± 1.39, *p* < 0.001) and group 3 (2.83 ± 0.55, *p* < 0.001) compared to group 1 (4.11 ± 1.85, *p* < 0.001). No significant correlation was demonstrated between the presence of the pathological visual uptake of [^18^F]FDG in a heart valve and the visual uptake of [^18^F]FDG in the oral cavity (*p* = 0.160) for the three groups.

In group 1, 13 patients underwent surgical valve replacement. The bacterial species on the valve was determined by culture after surgical valve replacement for 10 patients. The valve culture was positive for oral flora (*Streptococcus mitis, Heamophilus parainfluenzae)* in only two patients diagnosed with IEs. In group 3, two patients underwent surgical valve replacement. The valve culture was not positive for the oral flora in group 3 (Table 1).

Furthermore, in all three groups, the extra-cardiac visual [^18^F]FDG PET/CT findings were mainly noticed in the lymph nodes, mediastinum, and lungs. The prevalence of patients who had extra-cardiac findings in groups 1, 2, and 3 was 95%, 86%, and 90%, respectively. No significant difference between groups was observed for the extra-cardiac findings (*p* = 0.842).

For groups 1, 2, and 3, the visual [^18^F]FDG uptake scores, oral SUV_max_, extra-cardiac [^18^F]FDG PET/CT findings, and valve SUV_max_ are summarized in Table 3.

## 4. Discussion

In this retrospective study, no significant correlation was found between the visual [^18^F]FDG uptake scores or SUV_max_, the possible locations of oral inflammation and infection, and recent dental treatments.

Two previous studies found a correlation between the visual [^18^F]FDG uptake and oral inflammation/infection, periodontitis, and apical periodontitis [27,28]. In the Kito et al. (2012) study [28], periodontitis was scored by panoramic radiograph, which is not a reliable way to measure the severity and extension of this typical disease. Shimamoto et al. (2008) [27] used clinical measurements of periodontitis, resulting in a more reliable severity score. These measurements were performed by calibrated dentists, in contrast to our study in which dental records were used as a reference. This may have resulted in an underestimation of periodontal disease in our study. Furthermore, for both studies, it must be mentioned that it is technically impossible to accurately measure the severity and extension of apical periodontitis. Therefore, the reliability of the data related to apical periodontitis is questionable [29]. Panoramic radiographs were acquired to measure the severity of dental caries, periodontitis, and apical periodontitis [27,28]. However, intraoral bitewing radiography or a solo radiograph is superior to panoramic radiography in diagnosing the severity and extension of periodontitis and apical periodontitis [29].

The positive correlation between the visual [^18^F]FDG uptake scores and oral inflammation and infection in the previously mentioned studies could also be explained by the extension and severity of periodontitis in that particular study population, which could be more pronounced than our included patients who regularly visited a dentist [27,28].

In this study, we compared the visual [^18^F]FDG uptake scores and SUV_max_ to dental restorations performed within two years before [^18^F]FDG PET/CT imaging, and found no significant correlation. In contrast to our study, Kao et al. (2003) [29] reported incidental findings on the [^18^F]FDG uptake in patients with dental caries. A dental restoration is a filling of the mineral tooth loss due to caries and an indication of caries in the past. This may indicate possible inflammation of the tooth pulp. In our study, no dental caries were found, probably because the patients were regularly checked by dental professionals. The dental treatments performed in sextants with true-positive findings were 90% dental restorations. A possible explanation is that dental restorations are, for the most part, performed by dentists.

The normal physiologic uptake in the oral cavity could be an explanation for the false-positive results. In addition to the teeth, the oral cavity and oropharynx also consist of structures such as mucosal surfaces, salivary glands, muscle, lymphoid tissue, and brown fat, which is responsible for higher glucose uptake [30].

Regarding oral health status and IE, no difference in the dental health or amount or type of dental treatments before the diagnosis of IE was found between the groups diagnosed with IE and those not diagnosed with IE. Tubiania et al. (2017) [31] observed that, in her study of dental treatments, antibiotic prophylaxes, and IE in patients with prosthetic heart valves, invasive dental treatments may be associated with oral streptococcal IE. The prosthetic heart valve is recognized as a risk factor for IE [31]. Interestingly, 84% of group 1 had previously had a prosthetic heart valve, compared to only 32% of group 3. These data are in line with the literature, wherein for around 20% of patients diagnosed with IE oral microorganisms (commensals) are found on infected valves [4,5]. Bacteremia with oral microorganisms is a normal condition, and the number of microorganisms is higher when the infection and inflammation are more severe [32]. Therefore, one could logically assume that only acute dental problems are a preventable risk for IE.

The nature, duration, and incidence of bacteremia caused by routine oral behavior such as eating, tooth brushing, and dental procedures question the importance of and emphasis on endocarditis antibiotic prophylaxis. It is notably difficult to determine the origin and cause of bacteremia. Antibiotics reduce the severity and/or incidence of bacteremia. The European Society of Cardiology and American Heart Association recommend antibiotics prophylaxis for patients at the highest risk undergoing invasive dental procedures [24]. Patients considered at the highest risk of IE are “patients with previous IE, congenital heart disease and rheumatic heart disease and selected heart transplant recipients” [13,24,32]. Several studies have concluded that there is no evidence to suggest that antibiotics prophylaxis is effective or ineffective against IE for patients at the highest risk of IE undergoing invasive dental procedures [33,34,35,36,37]. Furthermore, in 2008 the National Institute for Health and Care Excellence guidelines stated that “antibiotic prophylaxis against IE is not recommended for people undergoing dental procedures” [38]. There is no consensus on the consequences of this guideline issued by the National Institute for Health and Care Excellence [38,39]. Our results indicated that patients diagnosed with IE did not have a higher prevalence of dental treatments. Furthermore, no difference in oral health was observed between the groups in our study. No correlation was found between dental status at the time of [^18^F]FDG PET/CT and the occurrence of endocarditis, nor between the antecedent of dental care procedures and the occurrence of endocarditis. This may suggest an indication for reducing the prescription of antibiotics prophylaxis.

Some limitations of our study should be addressed. The retrospective design increases the risk of bias. A total of 48 dentists, using their treatment strategy and documentation, supplied dental information (oral hygiene, regularity of dental visits, number of dental treatments, DMFT, and status praesens), which could have resulted in differences in the registration of the severity of dental problems. This is probably most significant for the screening of oral hygiene (plaque index, gingivitis) and periodontal disease, as different screening methods are applied, subject to wide inter-operator variability. Oral hygiene and periodontal disease are the important contributors to the inflammatory oral load.

Furthermore, [^18^F]FDG PET/CT scans can be influenced by scattering. For example, dental implants or amalgam dental restorations may cause beam hardening and scatter on low-dose CT, which may influence [^18^F]FDG PET images; therefore, both attenuated and non-attenuated PET scans were visually evaluated.

For future studies, a prospective design that includes the assessment of the severity and extension of periodontal health and acute dental inflammation just before the [^18^F]FDG PET/CT scan is warranted. Furthermore, a prospective multicenter study with patients receiving a dental check-up just before the [^18^F]FDG PET/CT scan is warranted if the modification of the antibiotic policy to prevent IE is considered. Another option could be the use of more specific tracers, since [^18^F]FDG is an aspecific tracer with a high physiological uptake in the oral cavity.

## 5. Conclusions

The results of this retrospective study suggest that there is no correlation between the [^18^F]FDG PET/CT uptake in the oral cavity and dental treatments or inflammation/infection reported by the patient’s dentist. Furthermore, no correlation between IE and actual oral health status was demonstrated. Based on this study, there is no causal relationship between oral pathogens and the risk of IE. Further studies with proper dental check-up just before the [^18^F]FDG PET/CT scan are warranted to investigate the relationship between oral inflammations/infections and IE.

## Figures and Tables

**Figure 1 diagnostics-10-00625-f001:**
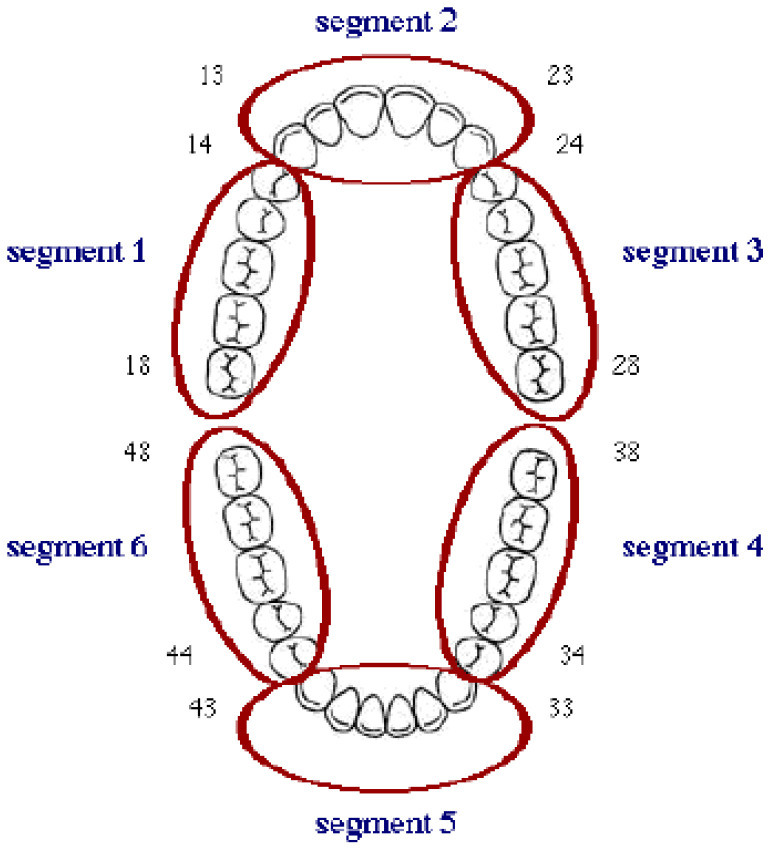
Oral cavity divided into six sextants. Sextant/segment 1: upper right premolar/molar region (18–14); Sextant/segment 2: upper jaw incisor/canine region (13–23); Sextant 3: upper left premolar/molar region (24–28); Sextant 4: lower left premolar/molar region (38–34); Sextant 5: lower jaw incisor/canine region (33–43); Sextant 6: lower right premolar/molar region (44–48).

**Figure 2 diagnostics-10-00625-f002:**
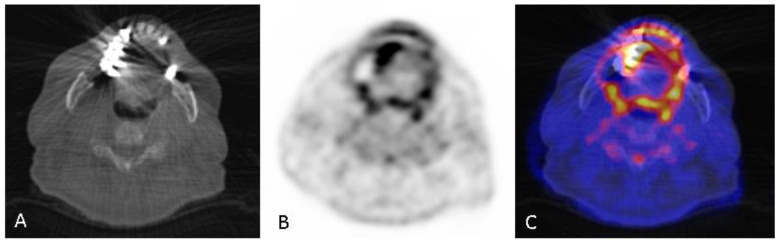
(**A**) a low dose CT on dental level acquired in combination with the PET. Seventy-eight-year-old male suspected of IE with a dental history of periapical inflammation related to tooth 16. Sextant 1 (**B**,**C**) shows a markedly increased [^18^F]FDG uptake (score 2, SUVmax 4.0), possibly due to the extraction of tooth 16. Sextant 3 (**B**,**C**) shows a markedly increased [^18^F]FDG uptake (score 3, SUVmax 7.7), possibly due to the extraction of tooth 25. Furthermore, periapical inflammation related to tooth 13 and tooth 27 was detected in panoramic radiographs but not in the PET scan. The patient had positive blood cultures for Streptococcus mitis.

**Table 1 diagnostics-10-00625-t001:** Patient characteristics and clinical data at the time of [^18^F]FDG PET/CT imaging.

	Group 1 (*n* = 19)	Group 2 (*n* = 14)	Group 3 (*n* = 19)
**Demographics**			
Age (median (IQR), years)	57 (42–66)	60 (37–65)	60 (51–70)
Male sex, *n* (%)	12 (63%)	7 (50%)	13 (68%)
BMI (mean ±SD, kg/m^2^)	25.1 ± 7.9	24.7 ± 2.7	25.5 ± 5.5
Diabetes, n (%)	3 (16%)	3 (21%)	4 (21%)
Previous IE, *n* (%)	5 (26%)	1 (7%)	4 (21%)
CIED, n (%)	2 (11%)	2 (14%)	3 (16%)
**Previous PHV ^○^, *n* (%)**	**16 (84%)**	**7 (50%)**	**6 (32%)**
Aortic	10 (63%)	6 (86%)	5 (83%)
Mitral	1 (6%)	1 (14%)	1 (17%)
Pulmonary	4 (25%)	0	0
Tricuspid	1 (6%)	0	0
**Echocardiography data available ^○^, *n* (%)**	**18 (95%)**	**14 (100%)**	**11 (58%)**
Vegetation ^○○^	12 (67%)	1 (7%)	0
Aortic	6 (50%)	0	0
Mitral	3 (25%)	0	0
Pulmonary	1 (8%)	1 (7%)	0
Tricuspid	2 (17%)	0	0
Abscess	2 (11%)	1 (7%)	0
Fistula	0	0	0
Prosthetic valve dehiscence	1 (6%)	0	0
Paravalvular leakage	6 (33%)	1 (7%)	0
CIED infection	1 (6%)	0	1 (10%)
**Microbiology data available ^○○○^, *n* (%)**	**19 (100%)**	**13 (93%)**	**13 (68%)**
Positive blood cultures ^○^	12 (63%)	8 (62%)	6 (46%)
*Staphylococcus aureus*	4 (33%)	6 (75%)	4 (67%)
*Staphylococcus lugdunesis*	0	0	1 (17%)
HACCEK ^++^	2 (17%)	0	0
*Streptococcus mitis*	3 (25%)	0	0
*Streptococcus bovis*	1 (8%)	0	0
*Streptococcus gallolyticus*	1 (8%)	1 (13%)	0
*Streptococcus angiosus*	0	0	1 (17%)
*Other ##*	1 (8%)	1 (13%)	0
*PCR and/or serology positive ^○^, n* (%)	**5 (26%)**	**0**	**0**
*Haemophilus parainfluenzae*	1 (5%)	0	0
*Tropheryma whipplei*	1 (5%)	0	0
*Enterobacter cloacae*	1 (5%)	0	0
*Escherichia coli*	1 (5%)	0	0
*Proprionibacterium acnes*	1 (5%)	0	0
*Valve culture, n (%)*	**10 (53%)**	**0**	**1 (5%)**
*Staphylococcus aureus*	2 (20%)	0	0
*Haemophilus parainfluenzae*	1 (10%)	0	0
*Streptococcus mitis*	1 (10%)	0	0
*Proprionibacterium acnes*	1 (10%)	0	1 (100%)
*No bacteria and yeast on the valve*	5 (50%)	0	0
*Lab test, n* (%)			
*CRP (median (IQR), mg/L)*	7.4 (2.8–46)	89 (36.3–148.5)	51 (19–145)
*Leucocytes (median (IQR), ×109/L*	8.2 (5.8–9.9)	10.1 (7–13.8)	9.6 (6.5-12)
*Treatment, n (%)*			
*Antibiotic therapy*	19 (100%)	6 (43%)	5 (26%)
*Days of antibiotic therapy (median (IQR)) ^^*	14 (8–38)	6 (6–9)	6 (2–8)
*Valve surgery replacement*	13 (68%)	0	2 (11%)

AV, aortic valve; BMI, body mass index; CIED, cardiovascular implantable electronic device; CRP, C-reactive protein; CT, computed tomography; [^18^F]FDG, 18F-fluorodeoxyglucose; IQR, interquartile range; MV, mitral valve; PCR, polymerase chain; PET, positron-emission tomography; PHV, prosthetic heart valve; PV, pulmonary valve; and TV, tricuspid valve. ^○^ Percentages of PHV, abnormal echocardiographic findings, positive blood cultures, and PCR/positive serology are relative to the number of available PHV, echocardiograms, and positive blood findings, and a positive PCR/serology. ^○○^ One patient had vegetation on both AV and TV. Another patient had vegetation on both AV and TV. ^○○○^ For blood cultures, PCR, and serology, even if they were no longer positive at the time of [^18^F]FDG PET/CT imaging, they were classified as positive if they had been positive during the clinical episode prior to the [^18^F]FDG PET/CT scan. ++ HACCEK: Haemophilus spp., Aggregatibacter- (Actinobacillus) actinomycetemcomitans (A. actinomycetemcomitans), Capnocytophaga spp., Cardiobacterium hominis, Eikenella corrodens, and Kingella bacteria. ## S. epidermidis, Gram-negative staph bacteria. ^^ For 8 patients, data about the days of antibiotic therapy before the [^18^F]FDG PET/CT scan could not be retrieved from the patient’s electronic files.

**Table 2 diagnostics-10-00625-t002:** Visual [18F]FDG uptake score and dental treatments.

	Visual [^18^F]FDG Uptake Score ^○^				
	0	1	2	3	*Total*
**Dental Restorations**					
0	185	17	30	11	243
1	33	6	6	3	48
2	14	0	2	1	17
3	0	0	1	0	1
4	3	0	0	0	3
*Total*	235	23	39	15	312
**Root Canal Treatment**					
0	230	23	39	15	307
1	5	0	0	0	5
*Total*	235	23	39	15	312
**Extractions**					
0	219	23	37	15	294
1	7	0	2	0	9
2	3	0	0	0	3
4	4	0	0	0	4
6	2	0	0	0	2
*Total*	235	23	39	15	312

^○^ Visual [^18^F]FDG uptake score, 0; no [^18^F]FDG uptake is visible, (1); mildly increased uptake (=muscle background), (2); moderately increased uptake (>muscle activity, <[^18^F]FDG brain activity), (3); ≥brain activity.

**Table 3 diagnostics-10-00625-t003:** [^18^F]FDG PET/CT scan in oral cavity and valve, and extra-cardiac findings.

	Visual Score [^18^F]FDG Uptake (Mean ± SD) ^○^	Oral SUV_max_ (Mean ± SD)	Extra-Cardiac Findings (MEAN ± SD) ^○○^	Valve SUV_max_ (mean ± SD) ^○○○^
**Group 1**	**0.51 ± 0.85**	**3.64 ± 2.18**	1.95 ± 1.43	4.11 ± 1.84
Sextant 1	0.53 ± 0.84	3.24 ± 0.59		
Sextant 2	0.47 ± 0.84	3.80 ± 2.09		
Sextant 3	0.89 ± 1.05	4.17 ± 4.01		
Sextant 4	0.32 ± 0.75	2.98 ± 1.29		
Sextant 5	0.53 ± 0.77	3.97 ± 1.46		
Sextant 6	0.32 ± 0.82	3.17 ± 1.77		
**Group 2**	**0.57 ± 1.01**	**3.65 ± 1.19**	2.00 ± 1.47	3.04 ± 1.38
Sextant 1	0.43 ± 0.85	3.13 ± 0.70		
Sextant 2	1.21 ± 1.31	4.50 ± 1.25		
Sextant 3	0.50 ± 1.02	3.77 ± 0.51		
Sextant 4	0.43 ± 0.85	3.05 ± 0.76		
Sextant 5	0.50 ± 0.94	2.80 ± 1.74		
Sextant 6	0.36 ± 0.93	4.70 ± 0.00		
**Group 3**	**0.35 ± 0.82**	**3.35 ± 1.08**	2.84 ± 1.61	2.83 ± 0.54
Sextant 1	0.42 ± 0.90	3.40 ± 0.94		
Sextant 2	0.32 ± 0.82	3.97 ± 1.21		
Sextant 3	0.32 ± 0.82	2.80 ± 1.21		
Sextant 4	0.74 ± 1.15	3.52 ± 1.38		
Sextant 5	0.21 ± 0.54	3.28 ± 0.26		
Sextant 6	0.11 ± 0.46	2.10 ± 0.00		

[^18^F]FDG, ^18^F-fluorodeoxyglucose; SD, standard deviation; Sextant 1, upper right premolar/molar region; Sextant 2, upper jaw incisor/canine region; Sextant 3, upper left premolar/molar region; Sextant 4, lower left premolar/molar region; Sextant 5, lower jaw incisor/canine region; Sextant 6, lower right premolar/molar region; SUVmax, maximum standardized uptake value. ^○^ Visual [^18^F]FDG uptake score, 0; no ^18^F-FDG uptake visible, 1; mildly increased uptake (=muscle background), 2; moderately increased uptake (>muscle activity, <[^18^F]FDG brain activity), 3; =>brain activity. ^○○^ Extra-cardiac findings: lungs, mediastinum, sternum, CIED, liver, pancreas, gastro-intestinal, lymphatic system, oropharynx, thyroid, musculoskeletal, abdominal, vascular, cutaneous, atrial, and sinuses. ^○○○^ Valve: aortic valve, mitral valve, tricuspid valve, and pulmonic valve.

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
