# Peer review of "Relationship between 18F-FDG Uptake in the Oral Cavity, Recent Dental Treatments, and Oral Inflammation or Infection: A Retrospective Study of Patients with Suspected Endocarditis"

_diagnostics, 2020, doi:10.3390/diagnostics10090625_

Round 1
Reviewer 1 Report
The manuscript submitted to Diagnostics entitled “Relationship between 18F-FDG uptake in the oral cavity, recent dental treatments, and oral inflammation or infection: A retrospective study of patients with suspected endocarditis” is an original article which aim to evaluate:
- correlation between [18F]FDG PET/CT findings, recent dental treatment, and affected oral cavity?;
- correlation between infective endocarditis (IE), oral health 24 status and (extra)cardiac findings on [18F]FDG PET/CT?
On my opinion the article is interesting, well written, with good English.
The authors examined [18F]FDG PET/CT scans of 52 patients. This study suggests that no correlation exists between PET findings in the oral cavity, and dental treatments or inflammation/infection. No correlation between IE and actual oral health status was demonstrated.
However, I highlighted some issues.
Abstract: The abbreviation for IE is missing. Are the results of this retrospective work sufficient to support authors' conclusions?
Introduction: Are there studies describing the use of PET to detect intraoral infection / inflammation? Better specify the objectives and methods of the study.
Methods: Why weren't other patients included in the control group?
Discussion. Are there other similar studies that have shown similar results? Did the authors find limitations in their study by comparing it with other in the literature?
Conclusions: Why other studies like this could elucidate about antibiotic policy for the prevention of IE? Some considerations are not supported by the results of this study and by the scientific literature on the subject (correlation between oral pathogens and IE). Please improve
After clarified the indicated points and revised the content of the article accordingly, I am available for a second round of peer review.
Author Response
Reviewer 1:
Comments and Suggestions for Authors
The manuscript submitted to Diagnostics entitled “Relationship between 18F-FDG uptake in the oral cavity, recent dental treatments, and oral inflammation or infection: A retrospective study of patients with suspected endocarditis” is an original article which aim to evaluate:
correlation between [18F]FDG PET/CT findings, recent dental treatment, and affected oral cavity?;
correlation between infective endocarditis (IE), oral health 24 status and (extra)cardiac findings on [18F]FDG PET/CT?
On my opinion the article is interesting, well written, with good English.
The authors examined [18F]FDG PET/CT scans of 52 patients. This study suggests that no correlation exists between PET findings in the oral cavity, and dental treatments or inflammation/infection. No correlation between IE and actual oral health status was demonstrated.
However, I highlighted some issues.
Abstract: The abbreviation for IE is missing. Are the results of this retrospective work sufficient to support authors' conclusions?
Reply/ we thank the reviewer for the helpful comments and we hope we addressed the suggestions well to further improve the paper.
Apologize for this. We added the abbreviation for IE. We added some more information to the conclusion: “..and extra-cardiac findings”.
Introduction: Are there studies describing the use of PET to detect intraoral infection / inflammation? Better specify the objectives and methods of the study.
Reply/ the reviewer is correct, indeed previous PET literature is described. We mentioned this by reference 18-20: “Since both types of bacteria are present in the oral cavity, [18F]FDG PET/CT can be used to detect periodontal disease and apical periodontitis (18-20)”. We hope this clarifies more the background of the current study.
Objectives and methods are better described now in section Introduction.
Reply/we adapted this , see also the final part of Introduction. “Therefore, this study aims, 1) to explore the correlation of [18F]FDG uptake in patients with recent dental treatments and/or inflammation and infection in the oral cavity; 2) the correlation between IE, oral health status and (extra)cardiac findings on [18F]FDG PET/CT.
Methods: Why weren't other patients included in the control group?
Reply/ Our control group was group 3 – patients without IE based on the modified Duke criteria. We used this group served as negative control.
Discussion. Are there other similar studies that have shown similar results? Did the authors find limitations in their study by comparing it with other in the literature?
Reply/ There are some studies in the literature with some similarities with our studies. In section Discussion we described two previous studies (ref 28-29). They found a correlation between visual [18F]FDG uptake and oral inflammation/infection, periodontitis, and apical periodontitis (27-28). In the Kito et al. (2012) study (28), periodontitis was scored by panoramic radiograph, which is not a reliable way to measure the severity and extension of this typical disease. Shimamoto et al. (2008) (27) used clinical measurements of periodontitis, resulting in a more reliable severity score.
Indeed we found also limitations of our study, such as the lack of measurements that were performed by calibrated dentists, as we included the dental records (line 9-10), due to the retrospective design of the study. Other limitations are described in the lower part of section Discussion (line 19-, last part Discussion).
Conclusions: Why other studies like this could elucidate about antibiotic policy for the prevention of IE? Some considerations are not supported by the results of this study and by the scientific literature on the subject (correlation between oral pathogens and IE). Please improve
Reply/ We agree with the reviewer the elucidation of the antibiotic policy for oral healthcare in the prevention of IE in the conclusion too much highlighted. We removed this part from the conclusion. The final conclusion is: “The results of this retrospective study suggest that there is no correlation between [18F]FDG PET/CT uptake in the oral cavity and dental treatments or inflammation/infection reported by the patient’s dentist. Furthermore, no correlation between IE and actual oral health status was demonstrated. Based on this study, there is no causal relationship between oral pathogens and risk of IE. Further studies with proper dental check-up just before the [18F]FDG PET/CT scan are warranted to investigate the relationship between oral inflammations/infections and IE.” We hope this will better state the message of our paper.
After clarified the indicated points and revised the content of the article accordingly, I am available for a second round of peer review.
Reply/ we thank the reviewer for her/his reviewing time.

Reviewer 2 Report
I have carefully read the manuscript entitled Relationship between 18F-FDG uptake in the oral cavity, recent dental treatments, and oral inflammation or infection: A retrospective study of patients with suspected endocarditis. The work concerns an important issue concerning the influence of the oral cavity condition on the circulatory system. The authors used Positron Emission Tomography - Computed Tomography (PET / CT) as the diagnostic tool in patients with suspected IE technique. As a result of the conducted research, they showed that no correlation exists between PET findings in the oral cavity, and dental treatments or inflammation / infection. In my opinion, the work deserves to be published in Diagnostics in its current form.
Author Response
Reviewer 2
I have carefully read the manuscript entitled Relationship between 18F-FDG uptake in the oral cavity, recent dental treatments, and oral inflammation or infection: A retrospective study of patients with suspected endocarditis. The work concerns an important issue concerning the influence of the oral cavity condition on the circulatory system. The authors used Positron Emission Tomography - Computed Tomography (PET / CT) as the diagnostic tool in patients with suspected IE technique. As a result of the conducted research, they showed that no correlation exists between PET findings in the oral cavity, and dental treatments or inflammation / infection. In my opinion, the work deserves to be published in Diagnostics in its current form.
Reply/ we thank the reviewer for the positive view on our paper and taking the time to review.

Round 2
Reviewer 1 Report
The authors have made substantial changes in the sections of the article that I had indicated. The manuscript may be accepted for publication, after positive comments from other reviewers.